# Optimize PLA/EVA Polymers Blend Compositional Coating for Next Generation Biodegradable Drug-Eluting Stents

**DOI:** 10.3390/polym14173547

**Published:** 2022-08-29

**Authors:** Naila Ishaque, Nauman Naseer, Muhammad Asad Abbas, Fatima Javed, Shehla Mushtaq, Nasir M. Ahmad, Muhammad Farhan Ali Khan, Naveed Ahmed, Abdelhamid Elaissari

**Affiliations:** 1Polymer Research Lab, School of Chemical and Material Engineering, National University of Sciences and Technology (NUST), H-12, Islamabad 44000, Pakistan; 2Bahria International Hospital, Department of Cardiology, Takbeer Block Sector B Bahria Town, Lahore 53720, Pakistan or; 3School of Natural Sciences, National University of Sciences and Technology (NUST), H-12, Islamabad 44000, Pakistan; 4Department of Pharmacy, Faculty of Biological Sciences, Quaid-i-Azam University, Islamabad 45320, Pakistan; 5Institute of Analytical Sciences, University Claude Bernard Lyon-1, CNRS, LAGEPP-UMR 5007, F-69622 Lyon, France

**Keywords:** cardiovascular disease (CVD), biodegradable drug eluting stents, polymeric blends

## Abstract

In this research work, polymer blends of poly-lactic acid (PLA)/ethylene vinyl acetate (EVA) were prepared as the drug carrier materials for a bi-layer drug-loaded coating film for coronary stents. Different optimum compositions of blends were prepared by using an intense mixer. Then, the blends were hot-pressed and later cold-pressed to prepare for films of different thickness. The changes in weight, surface analysis and biodegradability with increasing time were studied using Scanning electron microscopy (SEM), weight loss and biodegradability tests. The mechanical and thermal properties of drug-loaded films were studied through universal testing machine (UTM) and thermo-gravimetric analysis (TGA). The effects of PLA, EVA and drug contents on in-vitro drug contents were investigated through the Ultraviolet-Visible Spectroscopy (UV-VIS) chemical analysis technique. The results obtained clearly showed that the addition of PLA promoted the unleashing of the drug whereas the addition of EVA nearly did not have the same affect. The mechanical properties of these various films can be tuned by adjusting the contents of blend parts. The factors affecting the unleashing of the drug became a serious matter of concern in evaluating the performance of bio-resorbable drug eluting stents. As a result, today’s chemical blends may be useful drug carrier materials for drug-loaded tube coatings capable delivering purgative drug in an incredibly tunable and regulated manner.

## 1. Introduction

Stents are the efficient and preferable substitute for surgery that were first implemented in 1977 by Grüntzig, using balloon angioplasty (BA) [1]. The first human implantation of a self-expanding stent was reported by Sigwart et al. in 1987 [2] and the first human implantation of a balloon-expandable stent was reported in 1987 by Palmaz et al. [3], who continued this trend [4]. Stents are generally tubular implants that providemechanical strength to the stenotic arteries or other non-vascular ducts until the risk of complete closure is removed. There are two groups of stents: self-expanding and balloon-expandable [5]. Early stents were mostly made of metals (until very recently); hence, bare-metal stents constitute the first generation of stents. These continuous metal frameworks are comprised of alloys of stainless steel and cobalt-chrome (CoCr) for balloon-expandable stents and nickel-titanium alloys for self-expanding ones. Despite the fact that it was considered as a turning moment in the area of upsurge, the revolution was regarded as a turning point in the field of decline [6]. Their own disadvantages in the field of surgery were increased risk of thrombosis and restenosis. Intravascular injuries that lead to in-stent restenosis (ISR) are very likely to occur during stenting operations. ISR is the leading cause of artery blockage over time and stent failure is eventually caused by loss of artery patency [7]. Cascade occurrences lead to the loss of artery patency as follows: dysfunctional vascular endothelium creates ISR directly, and this begins to occur when anti-thrombic and anti-atherogenic characteristics are lacking [8]. The dysfunction of the artery in suppressing the proliferation of vascular smooth muscle cells (VSMCs) makes VSMCs overgrow into the blood vessel inwards that leads to ship overtime blockage [9]. As indicated by early studies (Fischman et al. 1994), re-intervention was needed in about 15–20 percent of all implanted stents patients within 6–12 months after the first ISR-related BMS implantation [5]. This issue has been addressed by all types of stents. Several types of bare metal stents (BMS) were produced during the following decade. Using BMS virtually eliminated many of the risks of sudden artery closing, but some restenosis remained; the rate of restenosis of treated arteries fell from 30% to 60% for balloon angioplasty to 10% to 30% for BMS [10]. A new era started with the first generation of drug-eluting stents (DES) with controlled release of sirolimus (CypherTM) or paclitaxel (TaxusTM) from robust polymers and significantly reduced angiographic and clinical restenosis levels relative to BMS [11]. It is known that all medical devices and treatments have drawbacks, but the ones identified with this innovative technology were rapidly recognized. Restenosis is still prevalent and very-late stent thrombosis is more frequent with DES of the first generation than with BMS owing to incomplete healing and re-endothelialization [12]. Although re-endothelialization is multifactorial, robust polymer surface coatings may play a role in the process [13]. In order to limit the sensitivity of the artery to the polymer, DES projects often seek to either dramatically reduce the amount of polymer on the stent or use biodegradable polymers [14]. Biodegradable polymer DES provides regulated elution of the active drug from the stent backbone by means of a biocompatible polymer coating that gradually degrades to inert organic monomers upon completion of its useful function, thus reducing the risk associated with the long-term existence of a robust polymer in the coronary vessel surface [15]. Though Byrne et. al. noticed no significant difference in the findings of three years between biodegradable polymer and permanent polymer DES, Holmes et, al. acknowledged the likelihood of biodegradable polymers enhancing the long-term protection of drug-eluting stents [16]. However, the potential benefits of biologically degradable polymers should be measured by a longer follow-up [17]. Where, for example, generation of 3D polymeric scaffolds loaded with an active compound by supercritical freeze extraction process was evaluated; this innovative process combines the advantages of the thermally induced phase separation process and of the supercritical carbon dioxide drying. In our research, we employed thermal processing without any solvent, so that the process is free from undesirable additives/reagents such as solvents, etc. Both thermal processing and freeze-drying processes have their own advantages to produce biomaterials [18]. In this context, design and selection of highly suitable drug and compatible materials as well as their compositions and processing are vital. Among the key factors to designing the composition of DES are the choice of an appropriate drug to be eluted in order to prevent undesirable health effects, suitable biopolymers that biodegrade to assist drug release under well-controlled kinetic model and the possibility of incorporating biocompatible but non-degradable polymers as drug carriers in order to optimize the release rate.

In consideration of above, for example, Aspirin is known as a salicylate and a nonsteroidal anti-inflammatory drug (NSAID) that is widely used for mild aches and pains as a pain reliever and for fever control [19]. It is also an anti-inflammatory medication and may be used as a blood-thinner [19]. People at a high risk of blood clots, stroke, and heart attack may use aspirin at low doses over the long term [20]. It is used in low doses to prevent the formation of blood clots and to reduce the risk of transient ischemic attack (TIA) and unstable angina [21]. Aspirin is also used in doctors with cardiovascular disease to avoid myocardial infarction, by stopping clot formation [22]. After percutaneous coronary interventions (PCIs), such as the placement of a coronary artery stent, a U.S. Agency for Healthcare Research and Quality guideline recommends that aspirin be taken indefinitely [23]. Frequently, Aspirin is combined with an ADP receptor inhibitor, such as clopidogrel, prasugrel, or ticagrelor, to prevent blood clots. This is called dual antiplatelet therapy (DAPT) [24].

For a biodegradable polymer to assist in drug delivery, Poly(lactic acid) (PLA) is widely used in many different biomedical applications due to its biocompatibility, complete biodegradability, and non-toxic degradation products. Polylactic acid, or polylactide (PLA), is a thermoplastic polyester with backbone formula (C_3_H_4_O_2_)*_n_* formally obtained by condensation of lactic acid C(CH_3_)(OH)HCOOH with loss of water. PLA has become a popular material as it is economically produced from renewable resources. Due to the chiral nature of lactic acid, several distinct forms of polylactide exist: poly-L-lactide (PLLA) is the product resulting from polymerization of L,L-lactide [25,26]. For medical implants, tissue engineering, orthopedic structures, drug delivery systems, PLA and its composites are widely used [27]. There is also poly(L-lactide-co-D,L-lactide) (PLDLLA)—used as PLDLLA/TCP scaffolds for bone engineering [28].

In order to develop optimum compositions for DES, various biocompatible non-biodegradable polymers have been employed. For this purpose, ethylene-vinyl acetate (EVA), also known as poly (ethylene-vinyl acetate) (PEVA), has been widely used in numerous studies [17]. It is transparent copolymers and its characteristics are dependent on the molar amount of monomers ethylene and vinyl acetate. The percentage of vinyl acetate usually varies from 10 to 40%, with the remainder being ethylene [29]. There are thus different types of EVA copolymer, which differ in the vinyl acetate (VA) content and the way the materials are used. EVA copolymer is a heat-processing substance which is versatile and inexpensive [30]. Due to its protection and biocompatibility, EVA has been used for an extended period of time as a biomaterial for artificial cardiac activity and for the distribution of drugs to treat thrombosis diseases [31]. Furthermore, EVA was also used as a medication courier for surgical implants, stents, and other drug delivery products to combat oral infections [31].

In consideration of above, the current research focusses on the versatile novel development of drug eluting stent (DES) that are coated with optimize compositions of Aspirin, biodegradable PLA and biocompatible non-biodegradable drug carrier and rate controlling EVA polymers. Various optimum blend compositions were processed without employing any solvent and through melt-processing in order to avoid any non-desirable residual solvent toxic effects. The role of PLA in releasing Aspirin upon biodegradation and EVA copolymers to assist in optimization of rate of release and effect on the mechanical properties has been explored. Various release models such as zero order, first order, Hixon Crowell and Korsmeyer Peppas model, and Higuchi model were applied to study mechanism of drug release. The optimum composition’s role to release drug within 48 h have been described and a prediction through the model to determine the drug release for a longer duration of time was made and also correlated with the reported literature.

## 2. Materials and Methods 

### 2.1. Materials

Ethylene vinyl acetate (vinyl acetate content = 26% (*w*/*w*), MFI = 5.5 g/min) was purchased from Asia Polymer Corporation (Taiwan). Poly (lactic acid) (Mw = 121.5 kDa, MFI = 6.09 g/10 min) was obtained from Nature Works (LLC, Minnetonka, MI, USA). The drug Aspirin (10 mg/tablet) was purchased from D. Watson (Islamabad, Pakistan). All of the chemicals used for the experimental work were of medical grades.

### 2.2. Preparation of PLA/EVA Blends

The blends were prepared by following melt mixing method using intense blend mixer. The mixer used consisted of two counter rotating triangle shaped rotors with a rotational speed of 1.25:1 and feed of 50 g. The mixer was pre-heated first for few min at 160 °C to attain the required temperature and then PLA and Aspirin were added followed by the addition of EVA. A total of 9 different blend compositions were prepared by varying different processing parameters which include temperature, mixing time and rotating speed of the mixer. The rotational speed of the mixer was initially set at 10 rpm and then increased to 70 rpm with varying temperature i.e., 180 °C, 165 °C, 155 °C, and 150 °C for sample 1, 2, 3, and 4, respectively. The processing conditions for the required blending process were fed into the HAAKE Polysoft software. Samples were cooled at room temperature without quenching. Out of these 9 prepared compositions, only the 4 best compositions were chosen for further processing based on their properties. The varying processing parameters and composition and are listed in Table 1.

### 2.3. Preparation of Sheets

As the result of blend formation, mushy gob-like material was obtained, which was then placed inside the hot presser preheated at 165 °C for few min and having pressure maintained at 100 bar. The mushy gobs were hot pressed for 5 to 10 min and then cold-pressed inside cold-presser for 5 to 10 min to get 0.5 mm sheets. The obtained films were then tested experimentally using different characterization techniques. On the basis of surface morphology, the 4 best films named as F-1, F-2, F-3 and F-4 were selected for further characterization. Based on best mechanical properties, PLA/EVA composition containing 80% PLA and 20% EVA namely sample F-1 was chosen for coating on stents using Dip Coating Method.

### 2.4. Characterization of Blends

Surface morphology and structure of all the blend sheet samples having thickness of 0.5 mm were investigated by SEM (JSM 6490LA). The samples were gold sputtered prior to their use for SEM analysis. The prepared samples were characterized by attenuated total reflection–Fourier transformed infrared spectrophotometer (FTIR-ATR) Bruker Model Alpha. The spectra were scanned over range of 400–4000 cm^−1^ and analyzed using the software named Essential FTIR to confirm the presence of required functional groups on the prepared films.

For studying the effect of increasing temperature on the physical and chemical properties, thermal analysis of the prepared samples was performed by using Thermal gravimetric analyzer. The thermograms covering the temperature range of 0–500 °C with 10 °C increase in temperature per min were analyzed. The mechanical properties of the samples such as percentage elongation, tensile strength, ductility, and brittleness of the prepared samples were tested by using universal testing machine (UTM) Shamizdu AX-20 Tensile testing graphs for the selected optimum blend sheets were prepared to check the ductility and elongation of the prepared samples.

For studying the in-vitro drug release profile of the prepared drug loaded films, the samples were cut into discs with a diameter of 1 cm and placed in tubes containing freshly prepared phosphate buffer solution (PBS) of pH 7.4. These tubes were then placed in an orbital shaker bath at a temperature of 37 °C ± 0.5 °C and a shaking speed of 75 rpm. At every predetermined time interval, the release medium was completely withdrawn and replaced by an equal volume of fresh PBS medium. The collected release medium was then subjected to Ultraviolet-visible (UV-vis) spectrometer JENWAY 7315 Spectrophotometer. The amount of drug concentration was calculated using the equation:(1)y=0.0302x+0.0146 

For the weight-loss test, the prepared 1 cm samples were withdrawn from the tubes after the time interval of 4 days. They were then gently washed with distilled water, wiped with filter paper and later dried in oven at 25 °C under vacuum for 48 h. Dry mass: *W*_1_ was measured immediately after drying of the samples and put in Equation (2) to get the final value of weight loss (*wt*.%).
(2)$$wt.\%\;=\;\frac{W_{o}-\;W_{1}}{W_{o}}\times \;100\%$$

For evaluation of release profile, several kinetic models such as first order, zero order, Higuchi and Korsmeyer Peppas’s were applied. 

## 3. Results and Discussion

### 3.1. Scanning Electron Microscopy (SEM)

For the morphological studies of prepared samples, the surface of the blend sheet samples was coated with a sputtered gold film prior to SEM analysis. In the case of low EVA content (samples F-1 and F-2), soft sheets showing full melting were prepared as shown in Figure 1a,b. When the EVA content increased (samples F-3 and F-4), the surface of the blend sheets appeared to be homogeneous, as shown in Figure 1c,d [32]. The reason for this homogeneity is the low mixing temperature of 150 °C whereas EVA melts at around 130 °C. Homogeneous structures characterized by cellular morphology were generated; the presence of drug is not detectable by SEM analysis and the results obtained put in evidence the fact that the drug did not interfere with the scaffolds formation [33]. Indeed, the composite structures generated are similar to those generated in a previous work, in which pure PLLA scaffolds were obtained by SFEP [18].

### 3.2. Fourier Transform Infrared (FTIR-ATR) Spectroscopy

FTIR-ATR analysis of all the samples was performed in order to confirm the formation of blends. Figure 2 represents the FTIR-ATR spectra of PLA/EVA polymeric blends recorded at room temperature with wavenumber ranging from 400 cm^−1^ to 4000 cm^−1^. The characteristic IR peak at wave number 1735 cm^−1^ represents C=O stretching of vinyl acetate and PLA and confirms the bond formation between them [34]. The peaks at 2920 cm^−1^ and 2850 cm^−1^ represents CH and CH_3_ stretching bond found in both PLA and EVA and the C-O stretching near 1182 cm^−1^ represents the functional group of PLA. The presence of these peaks confirms formation of PLA and EVA blends. However, due to increased quantity of EVA in films F-3 & F-4, less uniformity was observed [35].

### 3.3. Thermo-Gravimetric Analysis (TGA)

In order to study the thermal stability of the prepared samples, Thermal gravimetric analysis (TGA) of best composition (F1 & F2) chosen based on biodegradation analysis (explained in Section 3.4) was performed. Figure 3 represents the TGA curve of the samples, where the descending thermal curve indicates the occurrence of weight loss. A 14.3 mg sample of PLA/EVA film was analyzed from 0 °C to 500 °C at 10 °C per min within a nitrogen atmosphere at 30 mL/min. The decomposition process which takes place in the temperature regime of 300–350 °C is due to the intramolecular transesterification reaction (exchange of functional groups between esters and alcohols). The second stage of degradation can be observed in the temperature regime of 350–410 °C, which is due to the removal of acetic acid from the EVA co-polymer present in the blend [36]. In addition to this, the final stage of degradation observed for PLA–EVA blends in the temperature conditions ranging from 410 °C to 460 °C is mainly due to the escape of unsaturated butene as well as ethylene compounds in vapor form [37]. Thus, the decomposition in the region of 410 °C to 460 °C confirms the formation of PLA EVA blend. TGA test also shows that the blend is thermally stable till 300 °C and all the thermal changes are occurring after this temperature, which means that blend can perform its function in body without any thermal stresses or degradation [38]. Attaining thermal stability in the samples prove that an optimum composition in blends have been achieved that can be applied for coating purpose of a drug eluting stent.

### 3.4. Biodegradation Test

In order to check the biodegradability of the prepared PLA/EVA blends in the human body, the weight-loss tests of the prepared samples were performed. The samples for this test were prepared by taking 1 cm diameter of the samples and placing them in tubes containing freshly prepared phosphate buffer solution (PBS) of pH 7.4. These tubes were then placed in an orbital shaker bath at the temperature of 37 °C ± 0.5 °C and shaking speed of 75 rpm. The samples were taken out after time interval of every 0,4,8 and 16 days. They were then gently washed with distilled water, wiped with filter paper, and later dried in oven at 25 °C under vacuum for 48 h. The value of the weight loss is calculated by using Equation (1) and the data are given in the Figure 4. In Figure 4, sample F-1 with 80% PLA showed fastest degradation rate of 0.5%, indicating complete weight loss in around 2 yrs. Sample F-2 with 70% PLA showed degradation rate of 0.2% indicating complete weight loss in around 5 yrs. However the samples F-3 andF-4 with lower PLA content of 50% and 30% and higher EVA content of 50% and 70%, respectively, showed slowest degradation rate of almost less than 0.01%. This shows that PLA is degradable. The initial 4-day weight loss is due to a burst release mechanism where PLA absorbs water and burst along with releasing drug. According to this result, our optimum composition should contain EVA at less than 30% for drug eluting polymeric heart stents. Moreover, the biodegradation test has showed that with increase in the PLA content, the degradability increases, while EVA decreases the degradation rate [39].

### 3.5. Mechanical Testing

The mechanical testing of the prepared samples was performed to study the change in mechanical properties of the samples. The tensile test indicated a vast difference in the mechanical properties of pure PLA compared with PLA/ EVA blends. The strength of pure PLA is around 20 MPa with only 10% elongation at break and high elastic modulus [40]. In Figure 5, the tensile strength of sample F-1 decreased to 9.13 MP as compared to pure PLA UTS at 14.16 MPA while the % elongation was increased 160% from 12%. This implies that the addition of EVA increased the toughness of PLA, however, the load-bearing capacity decreased [41]. Sample F-2 showed drastic decrease in the tensile strength along with the elongation at break showed a remarkable increase as the EVA content was increased from 50%. The higher proportions of EVA suppressed the effect of PLA [42]. In the molecular structure of EVA, its polymer chains are highly branched into a coiled structure with a very high molecular weight because when it is blended with PLA it causes changes in the molecular structure of PLA from crystalline to amorphous [43]. This change in the molecular structure results in the decrease in crystallinity of PLA, which in turn decreases the load-bearing capacity or the tensile strength of the PLA/EVA blend. Moreover, EVA is an elastomeric, so it absorbs the impact energy and acts as stress concentrator [44]. It impedes the crack initiation and propagation due to which the toughness of PLA/EVA blend increases. The overall strength decreases with an increase in EVA content; however, the graphs slightly increase before the final crack due to a reason known as strain hardening [45], which is defined as the strengthening of a polymer at a large strain deformation. It is usually observed in polymers or elastomers such as EVA when they are stretched in plastic deformation beyond their yield point. Polymers that exhibit large amount of strain hardening are mostly tougher and undergo ductile formation. This degree of strain hardening for different polymers depends on both intrinsic and the extrinsic factors. In this case it depends on temperature, geometry and stress rate applied to blends [46]

### 3.6. In-Vitro Drug Release Study

An in vitro drug release study is an important tool for investigating and studying the drug release during the various stages of time. The values obtained from UV-vis spectroscopy were the subjected in Equation (1) to calculate the drug release content of the samples. Figure 6 shows the in vitro drug release curves of aspirin loaded film and aspirin coated stunt respectively. From the graph we can see that 1.34 µg of drug was released in 48 h. From other calculations, it was found out that 100% controlled drug release will take place in approximately 5 months (3582 h or 149.25 days), aligning with the reported literature.

By applying various release models such as zero order, first order, Hixon Crowell and Korsmeyer Peppas model, mechanism of drug release was determined as shown in Figure 7. R^2^ values indicate that release from follows Higuchi model. R^2^ value was 0.9976. 

## 4. Conclusions

The synthesis of PLA/EVA blends using the hot-melt intense mixing process was successfully carried out without the use of any chemical or non-chemical compatibilizer. In this research, blending with EVA succeeded our major concern of altering PLAs brittleness and elongation. Mechanical testing results showed the increase in EVA content decreased the brittleness and increased the % elongation. Sample F-1 and F-2 (15% and 30% EVA) provided the best results with strength of almost 10 MPa with decreased brittleness from 12 GPa to 4 GPa, followed by a significant increase in percentage of elongation from 10% to 160%. FTIR analysis confirmed the formation of PLA/EVA blends while SEM images for sample F-1(85% PLA and 15% EVA) showed smoothest surface without any visible traces of flakes or other particles compared with other compositions. The biodegradation test showed the fastest degradation for sample F-1 (85% PLA) at 2 years and slowest for sample F-4 with highest EVA content. TGA results indicated that the blends were stable till 300 °C, suggestsing that these blends can perform its function in the body without any thermal stresses or degradation. Thus, according to this research, the optimum PLA/EVA blend composition for biodegradable stents lies with EVA in the range of 15–30% along with PLA at 85–70%. 

## Figures and Tables

**Figure 1 polymers-14-03547-f001:**
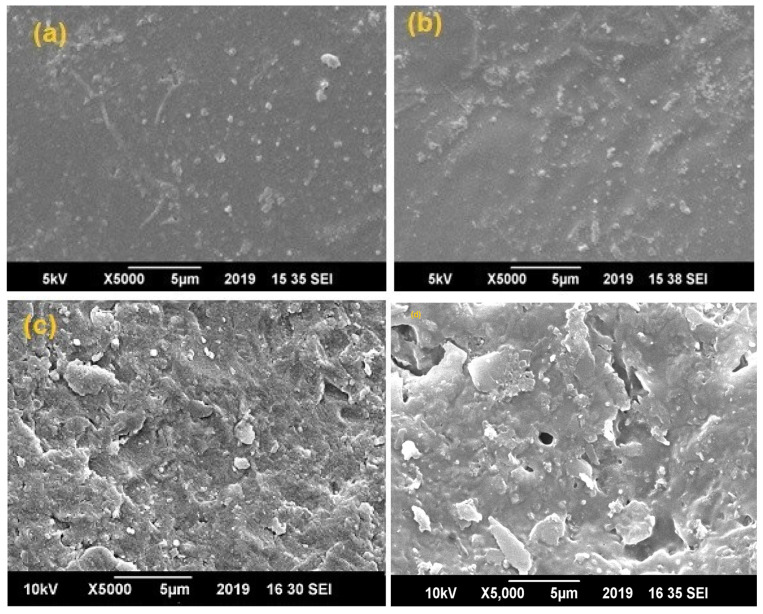
SEM images of (**a**) Sample F-1 (80% PLA and 20% EVA), (**b**) Sample F-2 (70% PLA and 30% EVA), (**c**) F-3(50% PLA and 50% EVA) and (**d**) Sample F-4 (30% PLA and 70% EVA).

**Figure 2 polymers-14-03547-f002:**
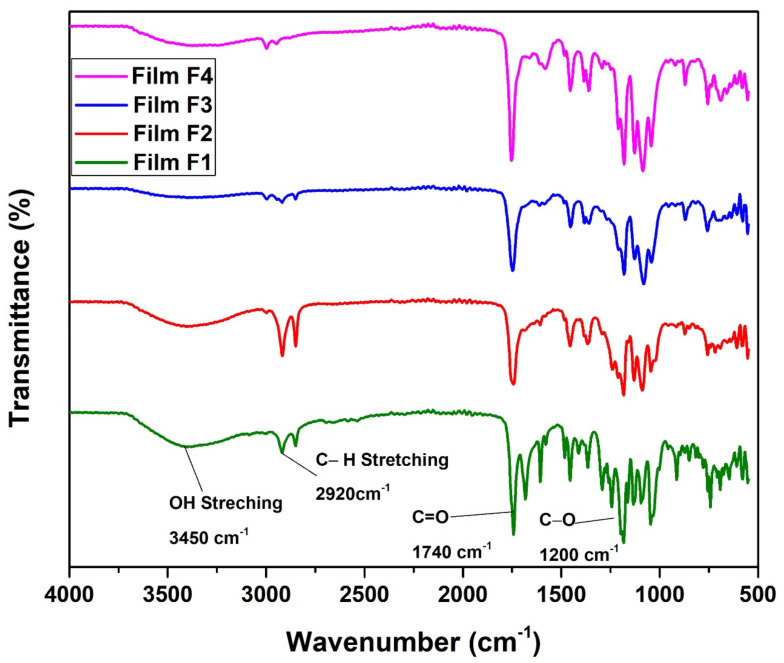
FTIR spectra for PLA/EVA blend compositions F-1 (80% PLA & 20%EVA) and F-2 (70% PLA & 30% EVA) F-3 (50% PLA & 50%EVA) and F-4 (30% PLA & 70% EVA).

**Figure 3 polymers-14-03547-f003:**
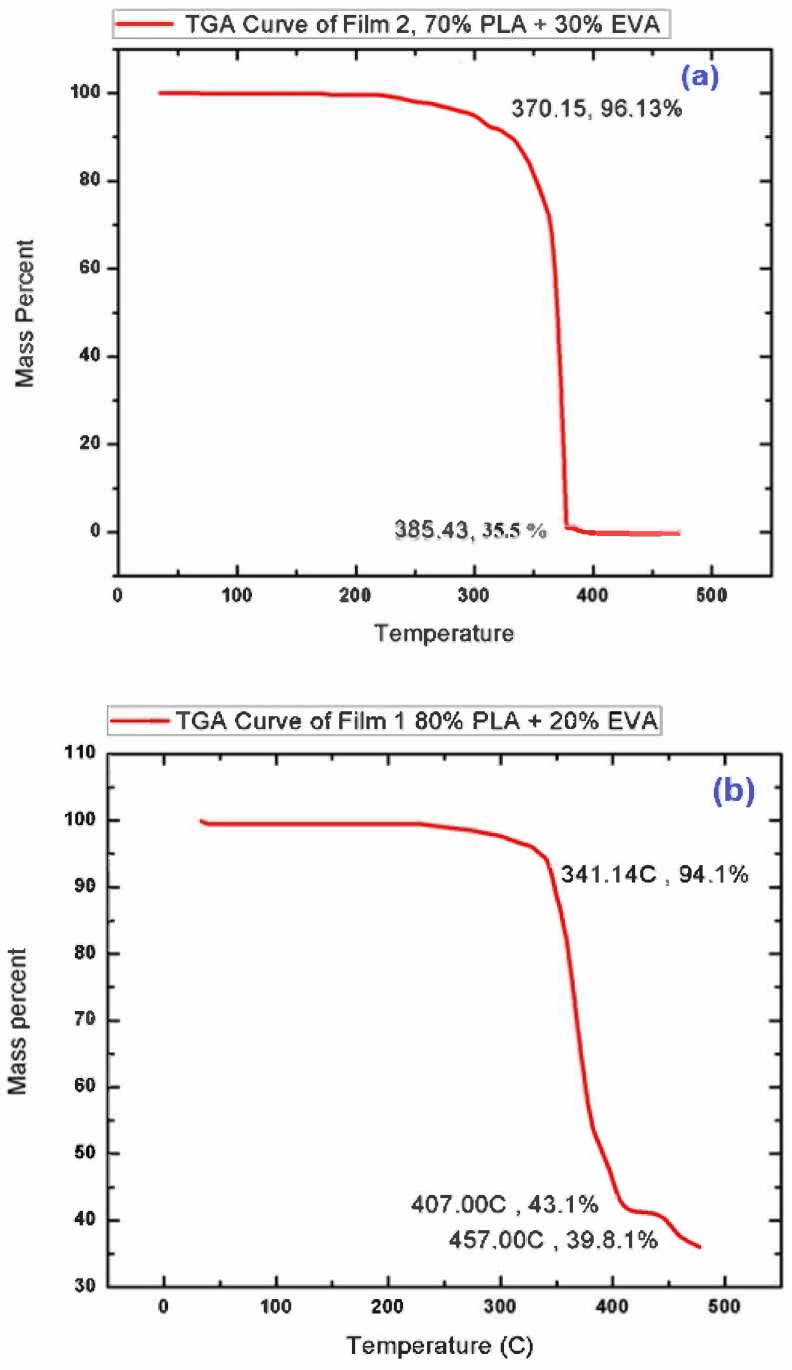
TGA curve of (**a**) F-1 (80% PLA & 20% EVA) and (**b**) F-2 (70% PLA & 30% EVA).

**Figure 4 polymers-14-03547-f004:**
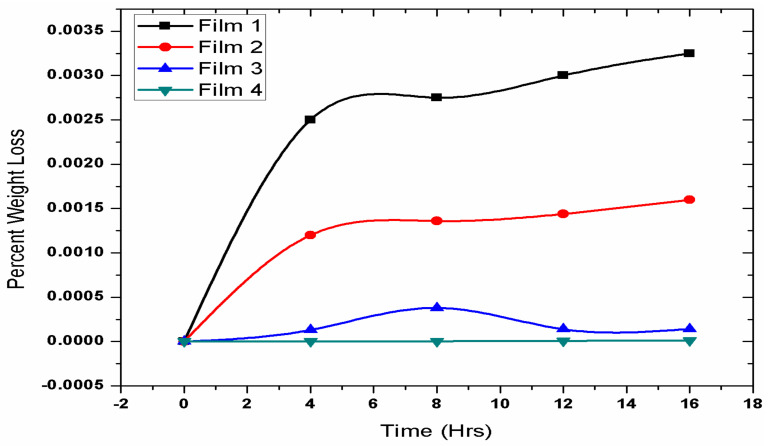
Biodegradation curves for F-1 (80% PLA and 20% EVA), F-2 (70% PLA and 30% EVA), F-3 (50% PLA and 50% EVA) and F-4 (30% PLA and 70% EVA).

**Figure 5 polymers-14-03547-f005:**
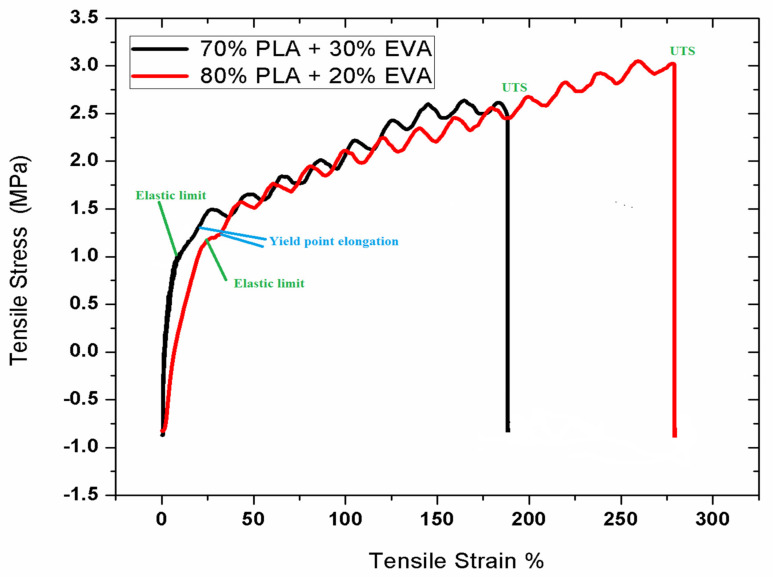
Stress vs. Strain graph for sample F-1 (80% PLA and 20% EVA) and F-2 (70% PLA and 30%EVA).

**Figure 6 polymers-14-03547-f006:**
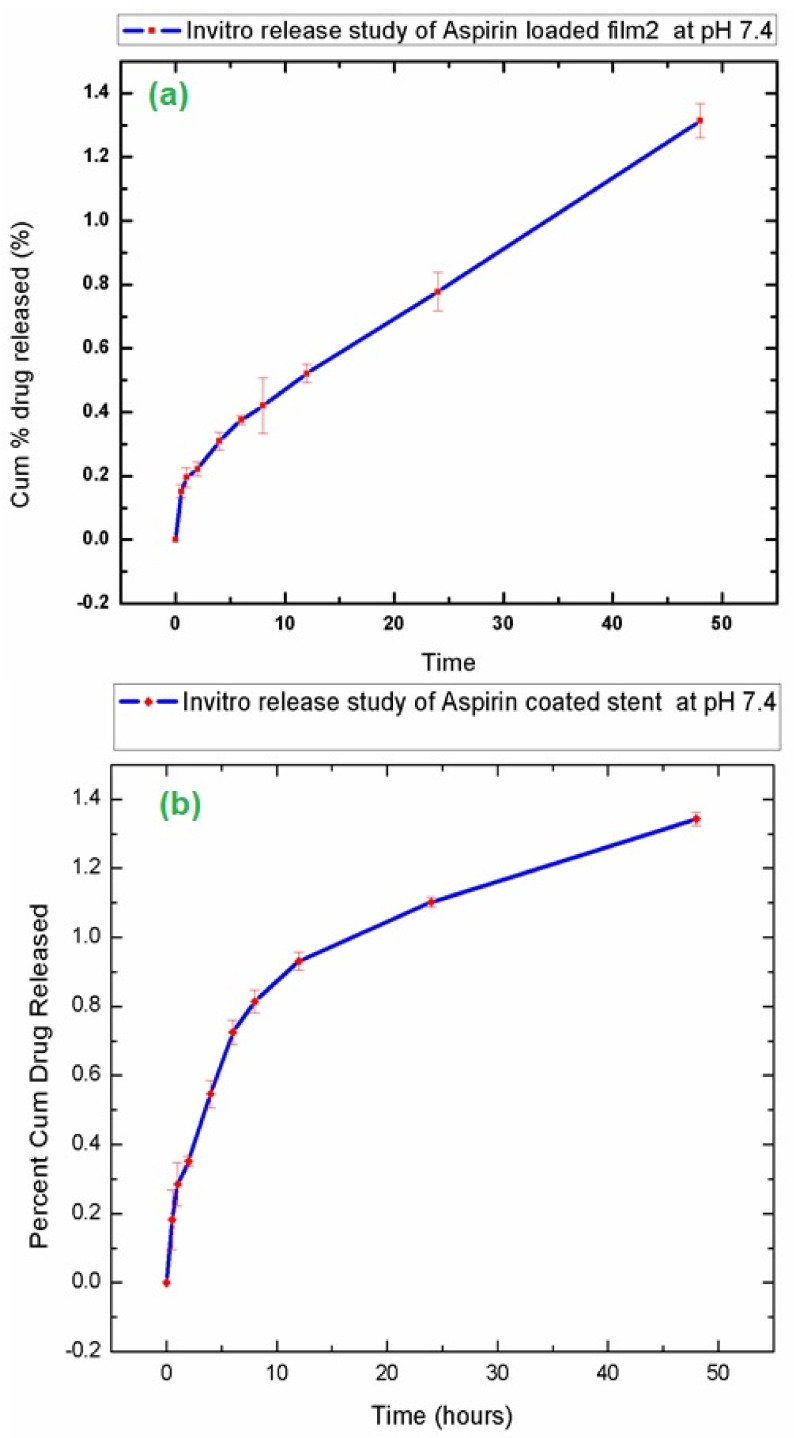
In vitro drug release study of (**a**) Aspirin loaded F-2 (80% PLA and 20%EVA) and (**b**) Aspirin coated stent.

**Figure 7 polymers-14-03547-f007:**
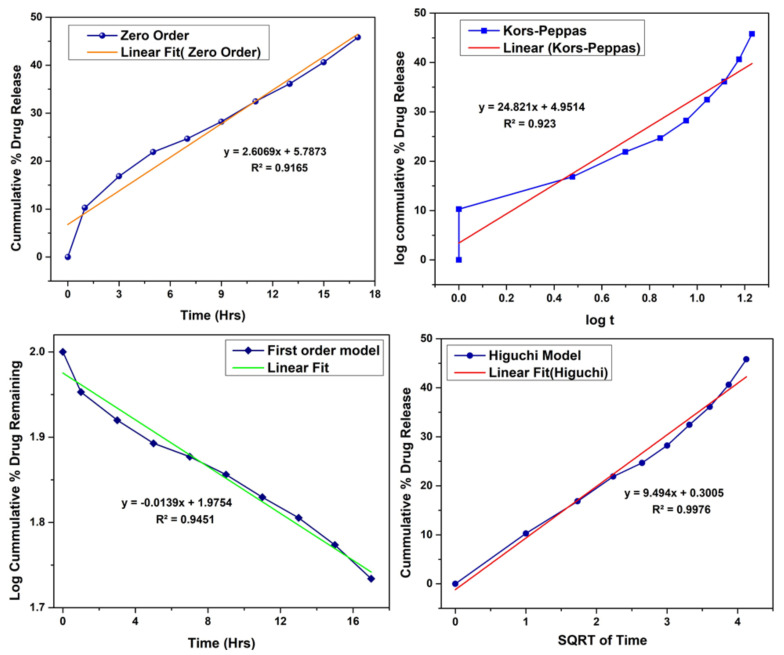
Drug release model kinetics.

**Table 1 polymers-14-03547-t001:** Processing parameters and composition of best-chosen samples.

Sample	PLA %	EVA %	Polymer	Drug %	Rotational Speed (rpm)	Temperature (T)	Time (t)
Film1 (F-1)	80	20	70	30	80	180	25 min
Film 2 (F-2)	70	30	70	30	75	165	25
Film 3 (F-3)	50	50	70	30	65	160	20
Film 4 (F-4)	30	70	70	30	50	150	20

## Data Availability

All the data is included in the article.

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
