# Peer review of "Optimize PLA/EVA Polymers Blend Compositional Coating for Next Generation Biodegradable Drug-Eluting Stents"

_polymers, 2022, doi:10.3390/polym14173547_

Round 1
Reviewer 1 Report
This paper reports on the “Optimize PLA/EVA Polymers Blend Compositional Coating for Next Generation Biodegradable Drug-Eluting Stents”. The article is interesting. Introduction is too short. Methodology, results and discussion can be improved. Reference seems be corrected.
I have few comments to the manuscript:
- Page 3 line 86. New chapter started “Poly(lactic acid) PLA, due…”
- Page 3 line 89. New chapter started “EVA copolymer…”
- Page 3. Fragments about ASA, PLA and EVA are too short.
- Page 3 line 91. Deleted extra space.
- On what basis was the proportion (composition) of the blend selected?
- The boiling point of ASA is 140 degrees Celsius.The article does not present the potential behavior of ASA at processing temperatures (150 - 180 degrees Celsius).TG for ASA is missing.This test will show that it makes sense to use middling at the processing temperatures used.
- No estimation of ASA losses in the processing steps.There is no discussion on this.
- Page 4 line 135. New chapter started “Thermal analysis…”
- Page 4 line 149 and 154. Deleted “…………………”.
- Page 5 line 178 and 179. Deleted extra space.
- Page 5 line 180. Deleted extra [28].
- Missing FTIR spectra of 100% PLA and EVA to compared.
- Was the effect of the addition of ASA to the tested films visible on the FTIR spectrum?
- On what basis were the films selected for TG testing?
- Why has no TG study been conducted for films filled with ASA?
- Why has biodegradation and mechanical properties not been conducted for films filled with ASA?
- Page 9 line 242. Add extra space.
- Page 9 line 246. Deleted extra space.
- “From other calculations it was found out that 100% controlled drug release will take place in 5 months approximately (3582 hours or 149.25 days) which is in good agreement with the reported literature.” Does this mean that the results were calculated on the basis of an initial guideline (two-day measurement).Without estimating the loss of ASA during production, the diminishing amount of agent release over time from the matrix?
- There are no citations from 2020-21 in the references
Taking into account all comments the manuscript may be published in Polymers after major revision.
Author Response
Dear Reviewer,
The response to your comments have been delayed due to the COVID-19 situation in the country. We addressed your suggesting in the best possible way . Hope you would find the manuscript quite satisfactory now

Reviewer 2 Report
The manuscript “Optimize PLA/EVA Polymers Blend Compositional Coating for Next Generation Biodegradable Drug Eluting Stents” deals with the production by a melt mixing method of drug eluting stents based on polylactic acid and ethyl vinyl acetate formulations. The work is interesting and well organized. However, some revisions are required before the publication.
In particular:
- Abstract. Please, define acronyms the first time they appear.
- Please delete “:” in the title of paragraphs.
- Introduction. The state of the art related to the production of biopolymeric devices for drug release, using alternative techniques, can be enlarged. For this purpose, please see this work: Cardea et al., 3D PLLA/Ibuprofen composite scaffolds obtained by a supercritical fluids assisted process, Journal of Materials Science: Materials in Medicine, 2014, 25(4), pp. 989–998.
- Please, add unit of measurement in the graphs of Figure 3.
- Graph in Figure 4 is not proportional in H x L. Please, check the correct dimension.
- Results. The morphological study performed by SEM is not clear in the results obtained. In particular, explain what kind of morphology is expected for this application and comment critically the results observed by SEM.
- Conclusions are a summary of the work. Please, rewrite in a more critical way, highlighting only the main results obtained and if they are satisfactory for the specific application.
- English should be improved.
- Some typing errors are present. Please, check and correct them.
Author Response
Dear Reviewer,
The response to your comments have been delayed due to COVID-19 situation in our country. We have tried to respond to the suggestion in best possible . Hope you will be satisfied by our response

Round 2
Reviewer 1 Report
The manuscript may be published as is.
Reviewer 2 Report
The revised version of the manuscript can be accepted in current form.